# The Effect of COVID-19 on Maternal Mental Health and Medical Support

**DOI:** 10.3390/medsci11010002

**Published:** 2022-12-23

**Authors:** Laura Catalina Merlano, Sindhu Nagarakanti, Kailyn Mitchell, Charles Wollmuth, Peter Magnusson, Joseph Pergolizzi

**Affiliations:** 1College of Medicine, Florida State University, Tallahassee, FL 32306, USA; 2Honors Program, NEMA Health Scholar, Temple University, Philadelphia, PA 19122, USA; 3Department of Physical Medicine, University of Nebraska Medical Center, Omaha, NE 68198, USA; 4NEMA Research Inc., Naples, FL 34108, USA; 5Department of Medicine, Cardiology Research Unit, Karolinska Institute, 17177 Stockholm, Sweden

**Keywords:** COVID-19, pregnancy, COVID-19 lockdowns, mental health

## Abstract

The Coronavirus disease 2019 (COVID-19) is a pandemic that affected the overall mental health of the population. As seen in previous situations, there seemed to be an extreme impact of disasters on the mental health of pregnant women and new mothers; therefore, we investigated the relationship between COVID-19 and maternal mental health. The pregnant subjects were identified during the study period through convenience sampling. The study received Institutional Review Board approval and online surveys were sent to subjects via email. The questions were focused on feelings about being pregnant and the influence of the practices during the pandemic. Fifty-one (51) pregnant patients were identified. Our study found that 92.3% of the participants felt negatively, as the COVID-19 precautions did not permit their significant other to attend their routine prenatal visits with them. 64.7% felt that the visits were less personal, 100% felt that they had to take more precautions. Only 42% of the doctors of the subjects discussed how COVID-19 could affect the pregnancy and the baby. Pregnant subjects all had negative feelings towards the pandemic, routine precautions, and the inability to include significant others in prenatal visits and delivery. The majority did not feel their medical teams discussed how COVID-19 could affect the baby.

## 1. Introduction

The Coronavirus disease 2019 (COVID-19) became a public health emergency in the United States after community transmission was first detected in February 2020 in both California and Washington State [1]. COVID-19 is a novel illness caused by a member of the *Coronavirida* family of viruses [2]. The virus, named severe acute respiratory syndrome coronavirus II (SARS-COV-II), shares 79.5% of its genome with the SARS coronavirus responsible for an outbreak in 2002–2003 in China that spread to 27 countries over an eight-month period [1]. However, SARS-COV-II has key differences compared to the original SARS virus; asymptomatic individuals can transmit the virus. The highly transmissible novel coronavirus quickly became a global pandemic, as declared by the World Health Organization on 11 March 2020 [1].

World-wide lockdowns and mandates restricting essential and non-essential services impacted many vulnerable populations during the COVID-19 pandemic, especially women in the perinatal period who require consistent medical monitoring. Health precautions and restrictions resulted in many women attending appointments alone, having appointments canceled, or utilizing virtual health visits instead of in office [3,4]. The stress of the pandemic coupled with the state of pregnancy may provoke mental health concerns from anxiety, depression, and fear due to social distancing measures and concerns over possible infection.

Previous data show that disasters can affect maternal mental health and perinatal health outcomes, which may demonstrate long-term consequences for children born during the COVID-19 pandemic [5]. This study aims to supplement growing data about the attitudes and practices of pregnant women during the initial outbreak of COVID-19 before vaccines were available to the public. It also aims to investigate how their prenatal experience was impacted by COVID-19.

## 2. Materials and Methods

### 2.1. Respondents and Procedure

The study was conducted on pregnant women. The inclusion criteria was pregnant women between the ages of 18 and 45. The exclusion criteria was anyone who is not pregnant and between those ages. The study was a survey and the convience sample was collected through social media, specifically through Facebook and Reddit groups on pregnancy. Some participants were recruited as acquaintances of the primary investigator. Participants were required to complete electronic informed consent before continuing with the questionnaire. Additional inclusion requirements included being pregnant between March 2020 to the moment of the survey. The survey was accessible via multi-devices and browsers. Online surveys were conducted between November 2020 through December 2020.

A questionnaire was completed by participants about how COVID-19 has affected their experience as expecting mothers to further assess their feelings towards being pregnant during a global pandemic and determine what current practices they were taking with regard to COVID-19. The questionnaire was developed by a team at the NEMA Research Group (Naples, Florida), with the goal of helping to quantify the experiences of these subjects during the pandemic. Additional inquiry included general demographics, medical appointment details (if partners or support persons were allowed in appointments, etc.). A total of 13 multiple choice questions, ranging from four to seven possible answers per question, were presented during the survey. Seven yes or no questions were included to provide general information about the pregnancy. Two fill-in-the blank questions were included to inform the questionnaire of the age and location of the subject.

Survey questions were presented as follows:By clicking “I agree” below you are indicating that you are between the age of 18 and 45 years old, have read and understood this consent form and agree to participate in this research study. Please print a copy of this page for your records.How old are you?What state do you currently reside?What is your highest level of education? If currently enrolled, highest degree received [multiple choice]Which of the following best describes you? [multiple choice]Where do you get your information regarding COVID-19? [multiple choice]Are/were you pregnant at any time during the timespan of March 2020-Present? [Y/N]Is this your first pregnancy? [Y/N]If No, how many times have you been pregnant? [multiple choice]If No, have routine appointments been affected due to difficulties of finding childcare? [multiple choice]Was/is your partner allowed to attend routine visits with you? [Y/N]If No, do you think it has negatively affected you? [multiple choice]Due to Healthcare staff being required to wear masks and protective gear, do/did you feel like your visits have been less personal? [multiple choice]Do you think COVID-19 has made you take more precautions during this pregnancy? [multiple choice]Have you already delivered your baby? [Y/N]If Yes, do you think the hospital rules of prohibiting visitors (family, friends, etc,) was positive for your postpartum? [multiple choice]If Yes, did your birthing plan change due to COVID? [multiple choice]If No, have you considered different birthing options due to COVID? [multiple choice]Has your doctor discussed how COVID-19 could affect your pregnancy/baby? [Y/N]Do you work in a job where you are at a higher risk of catching COVID-19? [Y/N]If Yes, have you considered a job change or working remotely because of the pregnancy? [multiple choice]Is/was your pregnancy considered high-risk? [Y/N]If Yes, do you think you have had support (from your doctor, spouse, family, etc.) in managing stress and concerns for your high-risk pregnancy during the COVID-19 pandemic? [multiple choice]Do you have any other concerns not mentioned or details about being pregnant during COVID-19 you’d like to discuss?

An independent internal review board (IRB) reviewed and approved the study prior to study initiation.

An email invitation to participate in the survey and/or the survey information and web address was sent to subjects. Participants completed an online survey through http://www.surveymonkey.com (accessed on 1 March 2020), a secure website that was provided after recruitment. Desktops, laptops, netbooks, Chromebooks, tablets, and/or mobile devices could be used to access the survey. Subjects were required to provide their consent prior to beginning the survey. Subjects were required to complete the survey in one sitting and could not change an answer once it was submitted or the subject left the survey.

### 2.2. Statistics

Summary statistics were produced for quantitative data: the population size (N), standard deviation (SD), coefficient of variation (CV%), minimum, median, mean, and maximum through Excel. A frequency table (showing N and %) was produced to summarize qualitative data.

## 3. Results

### 3.1. Baseline Characteristics

A total of 51 pregnant women signed an electronic informed consent form and qualified for inclusion based on the following two conditions: Electronic consent obtained prior to answering any survey related questions, women who were during the pandemic (March 2020–December 2020) between the ages of 18 to 45.

The mean age of participants was 28.9 ± 2.58 years, with most participants being in Florida (47.1% n = 24) and Kentucky (25.5% n = 13). Most of the participants were Caucasian (56.9% n = 29) with over 62.7% (n = 32) receiving a bachelor’s or master’s degree as their highest level of education. This information and more regarding the demographics of participants can be found in Table 1.

### 3.2. Knowledge

Study participants noted the information they acquired regarding COVID-19 came from various sources. The most popular source listed was the internet, where 58.8% of participants reported they received their information about COVID-19. Respondents were also questioned about if their doctors had discussed the effects of COVID-19 on their pregnancy, to which 58% responded no. For 60.8% of the participants, this was their first pregnancy, and the remaining participants reported had reported at least one other pregnancy. These data are further summarized in Table 2.

### 3.3. Attitudes and Practices

Survey questions were designed to gain further understanding of respondents’ attitudes towards practices and expectations regarding COVID-19 as well as understanding how participants responded to the COVID-19 pandemic and how this ultimately may have affected their mental health.

Out of 51 mothers who were pregnant during the timespan of March 2020 to Present, the majority, 76%, answered that their partner was not allowed to attend routine prenatal visits. Furthermore, 59% thought their partner not being able to visit had extremely to moderately affected them. Thirty-two respondents (63%) recorded that was a slight to no effect on visits being less personal (due to healthcare staff being required to wear masks and protective gear), and all respondents reported they had to take some increased measure of precautions during this pregnancy, 45% of respondents answered that they have had to take an extreme increase in precautions. Twenty-one (40%) of the mothers indicated that this was not their first pregnancy. When these twenty-one respondents were asked if their routine appointments have been affected due to difficulties finding childcare, 71% answered that there had been slight to no difficulties. Participants were also questioned about their risk of contracting COVID-19 based on occupation. Of the 60% of respondents who answered that they work in a high-risk job, 47% had moderate to extreme consideration towards a job change or working online. Of the participants who indicated their child has already been born at the time of the survey, 55% reported no positive or negative effects postpartum due to hospitals prohibiting visitors after delivery. Ten respondents indicated their pregnancy was considered a high-risk pregnancy, and 100% of those respondents answered that they had moderate to extreme support (from their doctor, partner, family, etc.) in managing stress and concerns for their high-risk pregnancy during the COVID-19 pandemic.

These data are further summarized in Table 3.

### 3.4. Discussion

This study aimed to supplement growing material about attitudes and practices of expecting mothers during the COVID-19 pandemic was conducted before vaccines were available. It also aimed to understand better how the pandemic changed their prenatal experience. This survey was administered between November 2020 through December 2020, during peak lockdowns.

Education about COVID-19 was not primarily given by the health staff many of the participants were interacting with. Over 58% of participants reported receiving their information about COVID-19 from the internet or other sources. More than half reported that their doctor did not discuss how COVID-19 would affect their pregnancy. This lack of communication and uncertainty has the potential to contribute to negative outcomes and further perpetuation of mental health symptoms for women experiencing a lack of education and reassurance from health care providers [6,7]. Participants relied heavily on online resources for information, more so than from their healthcare team.

Of the 39 participants who answered “no” to the question “Was your partner allowed to attend routine prenatal visits with you?” 58.9% recorded that this moderately to extremely affected them negatively. It has been demonstrated in the past, that women who lack partner support are at a greater risk for antenatal depression [8,9]. While this result only offers the immediate response to how this consequence made the participants feel, it may be the first telling sign of future frustrations and possible mental health complications as a result of continued lack of support during a such a vulnerable time in the prenatal period [10]. The majority of participants reported having given birth to their child by the time this survey was completed. Of the 32 participants who had reported childbirth before the survey was completed, 32.6% reported that hospitals prohibiting visitors after delivery did not positively affect them. This further speaks on the lack of a social support group during a critically emotional time during the perinatal period, which may negatively impact their postpartum mental health [5,8]. With many of the women describing negative feelings towards how many of their healthcare experiences were handled during the prenatal period, it is important to consider and possibly monitor women during this time for postpartum depressive symptoms. Postpartum depression is often a continuance of feelings and symptoms that manifest during the prenatal period [11,12]. It is important to understand and highlight the mental health risk observed by mothers during the postnatal period. Some data suggests that this period has the highest risk for illicit drug use and alcohol abuse [13] It is important to understand this relationship through future research so that proper interventions can be taken to improve outcomes for struggling mothers.

This study has a number of limitations. It was a small online survey and, as such, did not have a control group. Convenience sampling was used to recruit respondents who were self-selected. Many were answering questions about prior events which are not always remembered accurately. The web-based questionnaire is unable to be confirmed against medical records due to the anonymous nature, however, this style of data collection is widely used and accepted in epidemiological research [14]. Other limitations included a small sample of women from various states which may have differed from each other regarding healthcare practices. Respondents were also highly educated and primarily white or Caucasian suggesting a light bias towards more healthy individuals with better resources. Women with higher educational statuses tend to have fewer negative postnatal outcomes regarding mental health [11]. More information was needed on socio-economic background as well as any prior significant mental health history. Finally, this study only briefly explored a small window of time in which the pandemic was just beginning. Further studies are needed to understand how the rapidly changing health care management models and regulations may continue to impact women during the various stages of their pregnancies and what this means for the future generation of expectant mothers.

## 4. Conclusions

This web-based survey demonstrated the concerned attitudes and viewpoints from pregnant subjects during the pandemic. Some factors such as being unable to have partners included in health care visits and receiving information from online resources as opposed to health care staff may further implicate an already difficult prenatal period. This period is critical to the well-being of mothers and children, further studies are needed to better understand how these factors have effected long term outcomes.

## Figures and Tables

**Table 1 medsci-11-00002-t001:** Demographics Table.

Measure	Item	Count	Percentage (%)
Pregnancy Status	Women who were Pregnant (March 2020–December 2020)	51	100%
Age	18–45	51	100%
State of Residence	Florida	24	
	Georgia	1	47.06%
	Kentucky	13	1.96%
	Louisiana	1	25.49%
	Massachusetts	1	1.96%
	Michigan	1	1.96%
	Mississippi	1	1.96%
	New Jersey	1	1.96%
	North Carolina	1	1.96%
	Tennessee	2	1.96%
	Texas	2	3.92%
			3.92%
Highest Level of Education	High School Diploma/GED	9	17.65% 13.73% 37.25% 25.49% 5.88%
	Associates Degree	7	
	Bachelor’s Degree	19	
	Master’s Degree	13	
	Doctorate	3	
Race/Ethnicity	Black or African American	7	13.73%
	Asian	0	0%
	White or Caucasian	29	56.86%
	Native American or Alaskan Native	0	0%
	Hispanic or Latino	9	17.65%
	Multiracial or Biracial	6	11.76%
	Not Listed	0	0%
Number of Preganacies	Pregnant once	31	60.78%
	Pregnant twice	13	61.90%
	Pregnant three times	5	23.81%
	Pregnant four times	2	9.52%
	Pregnant more than four times	1	4.76%
Has the Child been Delivered at time of survey Completion?	Yes	32	62.75%
	No	19	37.25%
High Risk Occupation for Contracting COVID-19	Yes	30	60%
	No	20	40%
Is this Pregnancy High Risk	Yes	10	20%
	No	40	80%

**Table 2 medsci-11-00002-t002:** Knowledge Regarding COVID-19.

Question	Item	Count	Percentage (%)
Where do you primarily get your information regarding COVID-19?	Internet	30	58.82%
	Television	6	11.76%
	Newspapers	1	1.96%
	Social Media	8	15.69%
	Family and Friends	6	11.76%
Has your doctor discussed how COVID-19 could affect your pregnancy/baby	Yes	21	42%
	No	29	58%

**Table 3 medsci-11-00002-t003:** Attitudes and Practices.

Measure	Item	Count	Percentage (%)
Was your partner allowed to attend routine prenatal visits with you?	Yes	12	23.53%
	No	39	76.47%
If your Partner was not allowed to attend routine visits, do you think it has negatively affected you?	Extremely	5	12.82%
	Very	11	28.21%
	Moderately	7	17.95%
	Slightly	13	33.33%
	Not at all	3	7.69%
Due to Healthcare staff being required to wear masks and protective gear, do you feel your visits have been less personal?	Extremely	3	5.88%
	Very	6	11.76%
	Moderately	10	19.61%
	Slightly	14	27.45%
	Not at all	18	35.29%
Do you think COVID-19 has made you take more precautions this pregnancy?	Extremely	23	45.10%
	Very	15	29.41%
	Moderately	8	15.69%
	Slightly	5	9.80%
	Not at all	0	0%
If this is not your first child, have routine appointments been affected due to difficulties finding childcare?	Extremely	2	9.52%
	Very	2	9.52%
	Moderately	2	9.52%
	Slightly	3	14.29%
	Not at all	12	57.14%
If you work in a high risk environment for catching COVID-19, have you considered a job change or working remotely because of the pregnancy?	Extremely	7	23.33%
	Very	4	13.33%
	Moderately	3	10%
	Slightly	8	26.67%
	Not at all	8	26.67%
If you have already delivered your baby, do you think hospital rules prohibiting visitors was positive for you postpartum?	Extremely	2	6.45%
	Very	5	16.13%
	Moderately	7	22.58%
	Slightly	7	22.58%
	Not at all	10	32.26%
If your pregnancy was considered High risk, do you think you had support (from doctor, partner family, etc.) in managing stress and concerns for your high risk pregnancy during the COVID-19 pandemic?	Extremely	4	40%
	Very	3	30%
	Moderately	3	30%
	Slightly	0	0%
	Not at all	0	0%
If your baby is already born, did your birthing plan change due to COVID-19?	Extremely	4	12.90%
	Very	5	16.13%
	Moderately	6	19.35%
	Slightly	5	16.13%
	Not at all	11	35.28%
If your baby has not been born yet, have you considered different options due to COVID-19?	Extremely	2	10.53%
	Very	1	5.26%
	Moderately	3	15.79%
	Slightly	3	15.79%
	Not at all	10	52.63%
Has your doctor discussed how COVID-19 could affect your pregnancy/baby?	Yes	21	42%
	No	29	58%

## Data Availability

Data available on request due to restrictions eg privacy or ethical. The data presented in this study are available on request from the corresponding author. The data are not publicly available due to privacy concerns.

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
