# Peer review of "The Effect of COVID-19 on Maternal Mental Health and Medical Support"

_medsci, 2022, doi:10.3390/medsci11010002_

Round 1
Reviewer 1 Report
The author studied the impact of the COVID-19 pandemic on the mental health of pregnant subjects through a questionnaire survey. This study has a strong timeliness and has a certain guiding role for the government to formulate relevant policies. However, The shortcomings of the paper are obvious, that is, the scale of the survey is small, the questionnaire is simple, and there is a lack of comparative research. Therefore, I think that the current version of the article is not suitable for publication, and suggest the author to further improve the research before resubmitting.
Author Response
The reviewer claimed that the survey conducted was small, but agreed that the survey contained guiding information for future policies and is extremely relevant to the time period. We reviewed the manuscript and have reviewed the research.
Addressing the comments about how the survey size is small and the unsuitable difficulty of the questions asked will not be possible, as going back and reconducting the survey cannot happen at this stage of the publication process.
Reviewer 2 Report
Undoubtedly, the topic raised by the authors is important due to the lack of prior scientific knowledge about the effects of the pandemic (Covid-19) - and we do not have this knowledge for obvious reasons. (Unfortunately, knowledge of the short-term and long-term effects of other infectious diseases - on pregnant women and their offspring born during an epidemic - is very limited.)
Therefore, the research presented in the article gives a chance to gain an insight into this phenomenon. Although this is only the first step towards recognizing the negative effects of the experiences of pregnant women as a result of the pandemic and restrictions in the healthcare system related to the Covid-19 pandemic, the results of the studies conducted by the authors are worrying. They indicate (among others) (1) negative feelings of women related to the lack of satisfactory information on the risks to pregnancy and women's health - from doctors and medical teams, (2) limited support from intimate partners (husbands, fiancées, cohabitants ... .) as a result of restrictions in the health care system. Women's awareness of the risk of the virus for their health and life and for the child - with the perceived objective limitation of support from a close person and limited contact with the medical team - may be a factor increasing the risk of psychosocial disorders in a woman and her somatic condition. There are still possible negative consequences of these changes, not only for the child but also for the partner, and finally - for the whole family. And in a global perspective - for the organizational and financial burdens of society.
Due to the cognitive and social importance of the problem - the article is worth publishing.
These issues were taken into account by the authors, although they were somewhat "softened" or "obscured" by the presentation of other data collected through the survey.
I wonder if it is possible to group the discussed results differently - to emphasize the most interesting results - due to the purpose of the study?
It would be good to simplify the presentation of the survey and the process of collecting research material - in my opinion. Is it necessary to repeat the survey questions / items? Wouldn't it be worth considering specifying the answer options from which the surveyed women chose - even for the first question assuming the choice of one of many possible answers?
Basically two options (Y / N) or five options (Extremely; Very; Moderately; Slightly; Not at all) were asked.
It is a pity that no "open" questions (even one) were asked, in which the respondents could present their own thoughts on their experiences from this period. However, this would require a different approach to research and could increase the difficulty of analyzing the research material.
It is also worth considering some of the limitations of the research identified by the authors: first of all, the issue of the lack of a control group - probably there is such a possibility, or at least it was not there at the time of the study).
Author Response
Considering the model of the paper, changing the discussion seems like it would make the paper feel more disorganized. Leaving the questions in the paper ensures that the readers can go back to the exact question asked and understand the conclusions drawn better. I do not really see a lack of a control group in the paper.
I looked into grouping the discussion by importance, but that threw off the order of the paper as the rest of the paper is organized by the order that the questions appear in in the survey. Not including the questions was also considered, but that cannot happen as it makes the paper seem extremely unorganized. I am unable to go back and change the options of the survey. The limitations of the research are identified towards the end of the discussion section.
Reviewer 3 Report
Dear authors.
Thank you for the opportunity to review this paper and congratulations to the authors for your work.
The manuscript is about how COVID-19 affected the mental health of pregnant women.
I have some questions that I will pass on to you for an explanation, as I think they may help to complete your work.
Abstract Section
The abstract is well structured but exceeds the recommended maximum word count (200 words). You say that you have investigated the correlation between COVID-19 and maternal mental health. However, you have not conducted any statistical tests to prove this claim.
Keywords
Include any keywords that are more specific to the paper and that allow comparison with other similar papers.
Introduction section
The introduction is too short and provides little in the way of background. Too little reference support in this section. Better define the research questions.
2. Materials and Methods
2.1. Respondents and Procedure
Explain that sampling has been by convenience.
Specify the dates of sampling, as between November 2020 through December 2020 it could be 60 days or 20 days.
I would put the form at the end of the manuscript, as Supplementary Materials.
To indicate which browsers and devices were valid for the survey it is sufficient to put multi-devices and browsers. It is not necessary to explain them all.
2.2. Statistics
The software used for data analysis has not been mentioned.
The manuscript would be more complete and interesting if statistical inference tests had been carried out.
3. Results
3.1. Baseline Characteristics
Is there a selection bias caused by convenience sampling, as 75% of the respondents belong to only 2 states?
Homogenise table 1, as it has different colored cells.
3.2. Knowledge
Homogenise table 2, as it has different colored cells.
3.3. Attitudes and Practices
Homogenise table 3, as it has different colored cells.
3.4. Discussion
I would label this as a separate section.
The limitations are very well explained and are a sample of the biases that can affect the paper.
4. Conclusion
The implication was mentioned, but as I commented in the abstract section we have no statistical evidence to confirm this.
Others
Too little reference support ,
Author Information is not available.
I think the study needs to be more scientifically sound, it is currently only an exploratory study, but it can serve as a start for further, more detailed studies.
Regards
Author Response
I addressed the comment about the abstract being too long and corrected it to be under 200 words. I also looked into the comments regarding the introduction, the materials, the results, the keywords, the conclusion, and the references and amended them to the best of my abilities.
I fixed the part in the abstract that uses a statistical term and made it a more general term, in order not to suggest that statistics were used to analyze the results. I also made sure that the abstract was within the word count as well. I included a couple more keywords that are a little more specific. I went into the introduction section and highlighted the research question a little more. I was unable to find the dates of the survey, the supplementary materials section seemed like it would prompt the readers to answer the survey, and I changed the sentence so that it did not list all the browsers. I listed the software used and I cannot change the statistical analysis. The sample used also cannot be changed at this point but the bias is addressed in the discussion. Homogenizing the tables makes the tables look a little more confusing and has the reader getting lost. The discussion section according to the set up of the paper should appear like this. I cannot get more statistical analysis but I agree that this is an exploratory study.
Reviewer 4 Report
The present article deals with the effect of covid-19 on maternal mental health and medical support.
The paper is very well written throughout, although I feel that it might benefit from minor implementations.
Introduction
The introduction is short and concise, but includes the relevant background literature. Yet, it might be suggested to further expand it.
At p.1 the author refers to covid 19 as a respiratory syndrome. Although this was clear since the beginning, we have accumulated evidence that the virus does not impact the respiratory system. For example, de Sà et al. (2021) focus on pancreatic inflammation associated with infection, whereas Del Mele et al. (2021) deal with myocarditis and covid-19. Although establishing a causal link is difficult, it is important to not reduce covid-19 to respiratory syndromes only.
At the end of p.1, the authors suggest that there might be long-term consequences for children born during covid-19, but I feel that we do not have evidence to establish this, yet. Hence, I would suggest to the authors to rephrase this sentence.
Materials and methods
Why did the authors choose as an inclusion criteria 18-45? I understand that in most countries being 18yo provide the person with being legally an adult, but why limiting the sampling to people younger than 45 yo? Is there any theoretical reason for this?
At p. 3 the authors present the questions but not the multiple choice answers. I feel that the readers could benefit from being aware of what such multiple choices were. Further, there must be a mistake as the questionnaire is reported twice.
The last questions in the questionnaire was an open question? Why are answers to this question not reported and analysed via qualitative methods?
Results
The result section is clear but I think that limiting the results to mere descriptive statistic is a limitation. Why didn't the author run any inferential statistics? Didn't they have specific hypotheses that they could test? Just as an example, I think that it might be interesting to explore whether the answers changed in accordance to specific situations. For example, answers to the question "Do you work in a job where you are at risk of catching covid-19?" might somehow influence how respondents answered to the other questions
Discussions
The discussions are in general clear, but they lack concreteness. What are the practical implications of this study? How do the results of the present work inform future research and policies? This becomes particularly relevant considering that the restrictions toward covid-19 are being reduced globally, generally speaking.
References
de Sá, T. C., Soares, C., & Rocha, M. (2021). Acute pancreatitis and COVID-19: A literature review. World Journal of Gastrointestinal Surgery, 13(6), 574.
Mele, D., Flamigni, F., Rapezzi, C. et al. Myocarditis in COVID-19 patients: current problems. Intern Emerg Med 16, 1123–1129 (2021). https://doi.org/10.1007/s11739-021-02635-w
Author Response
I tried to address all of the corrections regarding the introduction and did my best to expand it. I also worked on expanding the rest of the article on a more minor scale.
I changed the introduction so that COVID is not limited to a respiratory illness. I do not completely see where in the introduction the implication that there would be harm to the children was conveyed. The reason for using that age range is that it is the most common age for child bearing. The questions and the option choices that were in the study are listed in the table further down in the paper. There were no inferential statistics done as there were no clear hypotheses being investigated. The questions that were asked cannot be changed at this point. The discussion does not include information about the future as the study is not concrete enough to make these generalizations.
Reviewer 5 Report
The paper reports a study aimed to investigate the attitudes and practices of pregnant women during the initial outbreak of COVID-19. The topic is relevant and has many practical implications. However, the study has some critical methodological problems:
- The sample is very small: only 51 women have been involved
- The sample is not homogenous: 31 women have already delivered the baby. The mental health issues during pregnancy and after delivering a baby could be very different
- Ten women indicated their pregnancy was considered a high-risk pregnancy; this is an entirely different experience from other pregnancy conditions
- It is not clear if the scales used to measure the women’s conditions and perceptions are validated
- A theoretical background to select the measures is lacking
- The aspects measured are very general
- The analyses are only descriptive and not very informative
Moreover, the paper should include more information:
- The introduction should describe the literature in the area more in-depth. For example, other studies about maternal mental health during and previously the pandemic should be discussed.
- The questionnaire should be presented in a more specific way. The authors should not report the questions list but should describe the scale used and the indicators considered linking the measures to theoretical concepts. Moreover, information about the scales reliability should be included. The list of questions can be included in the Appendix.
- The methods sections should be better distinguished: sample, tools, procedure, analysis
- The discussion should consider theoretical and practical implications more in-depth
Author Response
I tried to go back and address the study itself and make it less generalized when writing about it. Changing the study completely and the sample size is not going to be able to happen unfortunately.
The survey itself is unable to be changed so all comments addressing that are unable to be fixed. Both the materials section and the discussion were formatted according to the guidelines. I edited the introduction section to be more descriptive.
Round 2
Reviewer 3 Report
Dear authors, thank you very much for answering and adapting the manuscript to the questions raised in the previous review. Your explanations are very convincing and accepted.
Some comments:
If the inclusion criterion is age, it is not necessary to exclude those who do not belong to that age group.
According to the homogenization of the tables, it is better understood.
Perhaps, in conclusion, I would add that this study can serve as an exploratory study for a more complete one, with a larger sample, more states, and statistical inference. In this way, it would be more justified (this is a personal suggestion and it is up to the authors' judgment to accept it).
Congratulations .
Best regards.
Reviewer 5 Report
The article was hardly changed. I have a doubt that it is not the right version